# 🦓 ZEBRA: Towards Zero-Shot Cross-Subject Generalization for Universal Brain Visual Decoding

**Haonan Wang, Jingyu Lu, Hongrui Li, Xiaomeng Li**[*]
The Hong Kong University of Science and Technology
hwanggr@connect.ust.hk, eexmli@ust.hk

## Abstract

Recent advances in neural decoding have enabled the reconstruction of visual experiences from brain activity, positioning fMRI-to-image reconstruction as a promising bridge between neuroscience and computer vision. However, current methods predominantly rely on subject-specific models or require subject-specific fine-tuning, limiting their scalability and real-world applicability. In this work, we introduce ZEBRA, the first zero-shot brain visual decoding framework that eliminates the need for subject-specific adaptation. ZEBRA is built on the key insight that fMRI representations can be decomposed into subject-related and semantic-related components. By leveraging adversarial training, our method explicitly disentangles these components to isolate subject-invariant, semantic-specific representations. This disentanglement allows ZEBRA to generalize to unseen subjects without any additional fMRI data or retraining. Extensive experiments show that ZEBRA significantly outperforms zero-shot baselines and achieves performance comparable to fully finetuned models on several metrics. Our work represents a scalable and practical step toward universal neural decoding. Code and model weights are available at: https://github.com/xmed-lab/ZEBRA.

## 1 Introduction

The compelling connection between neural decoding and visual understanding has positioned fMRI-to-image reconstruction [1, 2, 3, 4, 5, 6, 7] at the forefront of computational neuroscience and computer vision. As a non-invasive method for observing activity in the brain's visual cortex, fMRI signals offer the intriguing possibility of reverse-engineering human perception—translating blood-oxygen-level-dependent (BOLD) responses into detailed visual reconstructions of what a person sees. This ability, often described as a "brain camera," marks a major shift, with potential applications in mental state interpretation [8] and advanced brain-computer interfaces [9, 10].

Despite remarkable progress in reconstructing images from individual brain data, the field faces a critical challenge: current models struggle to generalize across individuals. This limitation risks confining such breakthroughs to research labs rather than enabling real-world applications. Recent efforts [4, 6] have attempted to address this issue by developing cross-subject reconstruction through a two-step approach: first, *pretraining* a unified model on multi-subject data, followed by subject-specific *finetuning*, as illustrated in Fig. 1a. However, subject-specific finetuning imposes significant barriers to practical clinical use due to several limitations: (1) Clinicians and neuroscientists must still depend on AI experts to fine-tune models for each new patient. (2) The fine-tuning process is time-intensive, often taking around a day, which hampers real-time applications in brain-computer interfaces [9, 10] and neurorehabilitation [11]. (3) There is no universal feature space capable of learning neural representations across human subjects, restricting broader exploration in cognitive

---

[*]Corresponding author.

39th Conference on Neural Information Processing Systems (NeurIPS 2025).

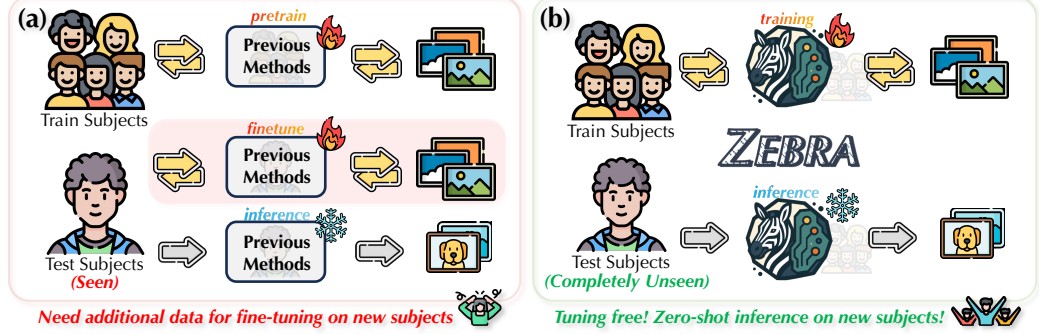

Figure 1: **(a)** Previous methods [4, 5, 6] typically involve two training stages: (1) pretraining a brain model with multiple subjects, and (2) fine-tuning the model for a specific subject. In this approach, the test subject is known to the model, which limits its zero-shot capability for new subjects. **(b)** In contrast, ZEBRA eliminates the fine-tuning stage, requiring training only once with the training subjects. This allows it to perform zero-shot inference on unseen subjects, achieving comparable performance to the fine-tuned approaches.

functions and variability. Thus, while developing **zero-shot cross-subject generalization methods** for brain visual decoding is critical, this area remains largely unexplored.

A straightforward approach would be to evaluate the previous state-of-the-art method, MindTuner [6], in a zero-shot setting and attempt to improve upon it. However, this is infeasible due to its subject-specific design, which is tailored to certain information content of representations across subjects. As a result, it fails when tested on a new subject whose representation carries a different amount of information. NeuroPictor [5] offers valuable insights by transforming fMRI data from different subjects into uniformly shaped 2D representations with spatial information preserved, facilitating the learning of a shared latent space. Nonetheless, its zero-shot performance remains limited since it is sensitive to subject noise, as shown in Fig. 6, and thus fails to learn invariant representations across subjects. Motivated by these observations, one may consider combining NeuroPictor's powerful unified brain encoding with the powerful decoding of MindTuner. Yet, even this possibly strongest baseline fails to achieve satisfactory results, as evidenced by the "Our baseline" and "NeuroPictor⋆" rows in Table 1.

To pave the way for zero-shot brain visual decoding, build on this baseline, we propose ZEBRA—**the first zero-shot brain visual decoding framework**—that enables direct generalization to unseen subjects without requiring additional fMRI data or model retraining (Fig. 1b). As illustrated in Fig. 2, the core idea of ZEBRA is to disentangle fMRI-derived features into four complementary components, with a focus on learning *subject-invariant* and *semantic-specific* representations. This design is motivated by neuroscientific evidence that, despite inter-individual variability in brain activity, the human cortex encodes semantic information in a consistent and topographically organized manner across subjects [12, 13, 14]. For a reconstruction framework to generalize effectively across individuals, it should preserve **subject-**

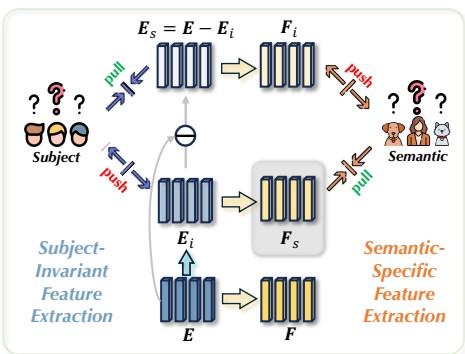

Figure 2: Core idea of ZEBRA. $F_s$ is used as diffusion prior guidance.

**invariant** (universal brain representations) and **semantic-specific** (class-discriminative) components, while suppressing subject-specific [15, 16] and semantically irrelevant variations. To this end, ZEBRA first extracts subject-invariant features by removing subject-specific noise via residual decomposition and adversarial training. In parallel, semantic-specific features are projected into a shared visual-semantic space and aligned with CLIP embeddings, ensuring semantic-level discriminability while remaining agnostic to subject identity. This disentanglement strategy enables robust cross-subject generalization and facilitates scalable fMRI-to-image decoding.

Extensive experiments validate the effectiveness of ZEBRA, particularly on low-level perceptual and pixel-wise metrics. ZEBRA achieves a substantial improvement in PixCorr, with a gain of +0.084 (0.153 vs. 0.069 of NeuroPictor), and an average improvement of +6.4 percentage points (81.8% vs. 75.4%) on Alex (5). More importantly, ZEBRA shows performance that is comparable to fully-finetuned methods in several metrics. For instance, it achieves an SSIM of 0.384, close to 0.375 of NeuroPictor (fully finetuned), despite not using any test subject data. Qualitative results further confirm that the visual reconstructions generated by ZEBRA are competitive with those produced by fully finetuned subject-specific models.

Our contributions can be summarized as follows: (1) We propose ZEBRA, the first zero-shot brain visual decoding framework that generalizes to unseen subjects without requiring additional fMRI data or finetuning; (2) We introduce a novel disentanglement strategy combining adversarial training and residual decomposition to learn subject-invariant and semantic-specific representations from fMRI signals; (3) We demonstrate that ZEBRA achieves competitive performance compared to few-shot and fully fine-tuned subject-specific methods across multiple quantitative and qualitative benchmarks.

## 2 Related Works

### 2.1 fMRI-to-Image Reconstruction

In recent years, with the rapid advancement and widespread adoption of functional magnetic resonance imaging (fMRI), researchers have increasingly recognized the critical value of fMRI signals for neuroscience research. Leveraging advanced technological tools, several datasets linking fMRI signals to images have emerged [17, 18, 19], with the Natural Scenes Dataset (NSD) being an exemplary instance [17]. Utilizing these datasets, the task of fMRI-to-image reconstruction has undergone significant development. Early research predominantly employed neural networks such as Variational Autoencoders (VAEs) and Generative Adversarial Networks (GANs) for reconstructing images from fMRI signals [20, 21, 22, 23]. However, these studies were either restricted to reconstructing simplistic digit images or lacked the capability to effectively represent both high-level and low-level semantic details clearly. The advent of diffusion models marked a transformative milestone, catalyzing numerous novel studies [4, 6, 24, 25, 26, 27, 5, 28]. Diffusion-based methods excel in capturing complex, high-dimensional semantic information. To effectively represent low-level spatial details, some approaches have incorporated blurry images as control inputs [4, 6], while others have utilized contours or masks [25, 28]. Moreover, certain studies have further refined high-dimensional semantic content by leveraging captions or labels [4, 6, 26]. By integrating both high-level semantic guidance and low-level spatial control, many recent approaches have produced intuitive and compelling reconstruction outcomes.

### 2.2 Cross-Subject fMRI-to-Image Reconstruction

In the medical domain, fMRI signals exhibit substantial variability across individuals due to differences in anatomical and physiological structures. Given the challenges associated with acquiring fMRI data, research focusing on cross-subject adaptation is profoundly significant. Central to cross-subject studies is the alignment of fMRI signals across different subjects. While anatomical alignment provides a foundational step, numerous studies emphasize the greater importance of functional alignment [29, 30] Numerous studies tried to address this issue [31, 32, 33, 34, 35, 36, 3]. Initial approaches typically trained separate models for each subject, resulting in poor generalizability and substantial data dependency issues [3]. Subsequent methodologies, such as MindBridge [31] and MindEye2 [4], sought to develop subject-agnostic networks by constructing shared latent space. However, these models often require extensive fine-tuning when encountering new subjects. While MindTuner [6] partially mitigates this requirement, it does not fundamentally resolve the persistent reliance on subject-specific data in cross-subject fMRI research.

## 3 Method

As illustrated in Fig. 3, based on a baseline with a ViT-based brain encoding backbone and a unCLIP generative model §3.1, ZEBRA improves this with two main components: (1) a Subject-Invariant Feature Extraction module that maps brain visual representations to shared latent space by exploring

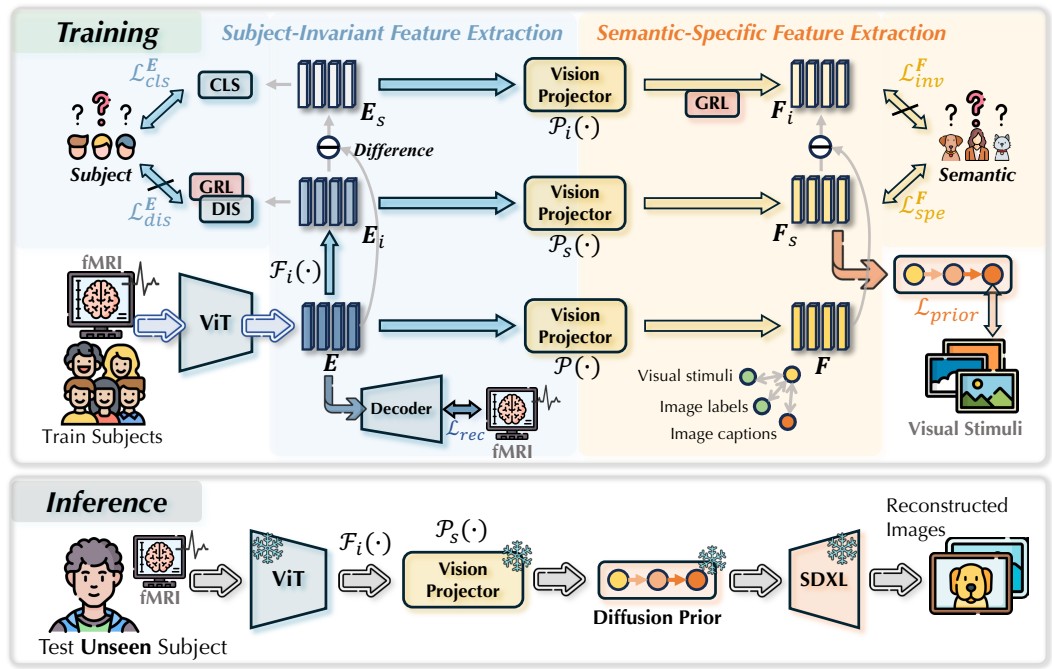

Figure 3: ZEBRA consists of two key components: (1) **Subject-Invariant Feature Extraction**, which disentangles subject-invariant representations from brain activity using adversarial learning and residual decomposition (§3.2); and (2) **Semantic-Specific Feature Extraction**, which aligns semantic information in brain features with vision-language embeddings via supervised learning and gradient reversal (§3.3). During inference, only the invariant projection path is used, enabling zero-shot generalization to unseen subjects.

subject-invariant features (§3.2), and (2) a Semantic-Specific Feature Extraction module that injects semantic features into the shared latent space. (§3.3).

## 3.1 Baseline Framework

Our baseline framework primarily consists of a ViT-based fMRI encoder (fMRI-PTE [35], pretrained on the UK Biobank dataset [37]) that maps fMRI data from different subjects into a shared latent space, and a diffusion prior network that converts the latent brain embeddings into vision features for image generation using Stable Diffusion. Given an fMRI scan and its corresponding visual stimulus $y$, we first transform the fMRI data into a unified 2D brain activation map [35], resulting in a single-channel image $x \in \mathbb{R}^{256 \times 256}$. The fMRI encoder then converts this 2D surface map into a latent representation $\boldsymbol{E} \in \mathbb{R}^{B \times L \times C_1}$, where $B$ is the batch size, $L$ is the number of tokens, and $C_1$ is the brain feature dimension. Then the latent representation $\boldsymbol{E}$ is converted to embeddings in CLIP space $\boldsymbol{F} \in \mathbb{R}^{B \times L \times C_2}$ as guidance for reconstruction. Following MindEye2 [4], we utilize a diffusion prior [38] to transform the fMRI-CLIP embedding $\boldsymbol{F}$ into a reconstructed OpenCLIP vision embedding $\boldsymbol{F}_y$ of the corresponding visual stimulus. Similar to DALL·E 2 [38], the diffusion prior is trained to minimize the mean squared error (MSE) between predicted and target embeddings:

$$\mathcal{L}_{prior} = \mathbb{E}_{\boldsymbol{F}_y, \boldsymbol{F}\epsilon \sim \mathcal{N}(0,1)} \left\| \epsilon(\boldsymbol{F}) - \boldsymbol{F}_y \right\|^2 . \tag{1}$$

The training of the baseline model involves three losses: (1) a contrastive loss $\mathcal{L}_{\mathrm{CLIP_t}}$ between the predicted CLIP text embedding $\boldsymbol{F}^t$ and the ground truth $\boldsymbol{F}_y^t$; (2) a contrastive loss $\mathcal{L}_{\mathrm{CLIP_v}}$ between the predicted CLIP vision embedding $\boldsymbol{F} \in \mathbb{R}^{B \times N \times C}$ and the ground truth $\boldsymbol{F}_y^v$; and (3) the diffusion prior loss $\mathcal{L}_{prior}$. Both $\mathcal{L}_{\mathrm{CLIP_t}}$ and $\mathcal{L}_{\mathrm{CLIP_v}}$ adopt the BiMixCo loss, which aligns video frames and corresponding fMRI signals using a bidirectional contrastive objective and MixCo-based data augmentation, detailed in the Supplementary.

## 3.2 Subject-Invariant Feature Extraction

**Residual Decomposition & Adversarial Training.** fMRI signals are highly idiosyncratic across individuals, making direct modeling challenging. To enable generalization, it is essential to filter out subject-specific noise and retain only invariant, stimulus-relevant components. The goal of *Subject-Invariant Feature Extraction (SIFE)* is to disentangle general brain representations into two components: subject-invariant features and subject-specific features. This disentanglement is achieved via *residual decomposition*, where we employ self-attention blocks $\mathcal{F}_i(\cdot)$ as the invariant feature extractor to derive subject-invariant features: $\boldsymbol{E}_i = \mathcal{F}_i(\boldsymbol{E})$. The choice of self-attention is not essential to enforcing invariance—which is primarily driven by the gradient reversal layer (GRL) and the associated adversarial losses—but serves to maintain architectural consistency with the brain encoder based on ViT. Subsequently, the subject-specific features are obtained as the residual difference: $\boldsymbol{E}_s = \boldsymbol{E} - \boldsymbol{E}_i$. This residual decomposition ensures that the extracted $\boldsymbol{E}_i$ captures components common across subjects, while $\boldsymbol{E}_s$ accounts for individual variability. The decomposition is further regularized by adversarial and supervised objectives to enforce disentanglement constraints and semantic consistency. To ensure that $\boldsymbol{E}_i$ is truly invariant to subject identity, we apply an adversarial training strategy [39]. Specifically, a subject discriminator $\mathcal{D}_{dis}$ is trained to maximize its ability to predict the subject label from $\boldsymbol{E}_i$, while the invariant extractor $\mathcal{F}_i$ aims to produce features that prevent $\mathcal{D}_{dis}$ from correctly identifying the subject. This adversarial objective is formulated as the following min-max game:

$$\min_{\theta_{\mathcal{E}}, \theta_{\mathcal{F}}} \max_{\theta_{\mathcal{D}_{dis}}} \left\{ \mathcal{L}_{dis}^{\boldsymbol{E}} := -\mathbb{E}_{x,s \sim \mathcal{X}, \mathcal{S}} \left[ s \log \mathcal{D}_{dis}(\mathcal{E}(\mathcal{F}_i(\boldsymbol{E}))) \right] \right\}, \tag{2}$$

where $s$ denotes the subject label. To guide the learning of subject-specific features $\boldsymbol{E}_s$, we introduce a subject classifier $\mathcal{D}_{cls}$, which is trained to predict the subject identity from $\boldsymbol{E}_s$. Its parameters $\theta_{\mathcal{D}_{cls}}$ are optimized via the following classification loss:

$$\min_{\theta_{\mathcal{D}_{cls}}} \left\{ \mathcal{L}_{cls}^{\boldsymbol{E}} := -\mathbb{E}_{x,s \sim \mathcal{X}, \mathcal{S}} \left[ s \log \mathcal{D}_{cls}(\boldsymbol{E}_s) \right] \right\}. \tag{3}$$

The residual decomposition mechanism inherently reduces the subject-specific signal in $\boldsymbol{E}_i$, thereby promoting the inclusion of subject-invariant content. We assume that subject-irrelevant features largely overlap with shared semantic representations. Thus, enforcing $\boldsymbol{E}_i$ to be adversarially invariant ensures that it encodes generalizable brain features across individuals.

**Representation Preservation Anchor.** While adversarial training promotes subject-invariant representation learning, it may inadvertently distort the original high-dimensional brain feature space $\boldsymbol{E}$. To counteract this, we introduce a *representation preservation anchor* via an auxiliary fMRI reconstruction task. Specifically, we employ a masked decoder $\mathcal{D}_{rec}(\cdot)$—comprising two deconvolution layers followed by a linear prediction head—to reconstruct the input signal as $\hat{x} = \mathcal{D}_{rec}(\boldsymbol{E})$. We adopt the mean absolute error (MAE) as the reconstruction loss, defined as:

$$\mathcal{L}_{rec} = \mathbb{E}_{(x,\hat{x}) \sim \mathcal{X}} \left[ ||\hat{x} - x|| \right]. \tag{4}$$

This reconstruction task serves as an anchor to preserve essential neural information in the latent space, ensuring $\boldsymbol{E}$ retains both biological fidelity and semantic coherence under adversarial training.

## 3.3 Semantic-Specific Feature Extraction

Given the subject-invariant brain representation $\boldsymbol{E}_i$ from SIFE, we further inject semantic information from stimuli to enhance semantic-specific alignment. Similar to SIFE, the *Semantic-Specific Feature Extraction (SSFE)* module consists of two components: Adversarial Training and a Representation Preservation Anchor.

**Adversarial Training.** We project brain features into the CLIP vision space using vision projectors composed of three linear layers with GELU activation [40], yielding three types of CLIP-aligned embeddings: semantic-specific features $\boldsymbol{F}_s = \mathcal{P}_s(\boldsymbol{E}_i)$, semantic-invariant features $\boldsymbol{F}_i = \mathcal{P}_i(\boldsymbol{E}_s)$, and general representations $\boldsymbol{F} = \mathcal{P}(\boldsymbol{E})$. The disentanglement between $\boldsymbol{F}_s$ and $\boldsymbol{F}_i$ is driven by the residual decomposition in the brain feature space. To ensure $\boldsymbol{F}_s$ captures meaningful semantic information, we directly align it with OpenCLIP vision embeddings $\boldsymbol{F}_y$, using the BiMixCo loss $\mathcal{L}_{spe}^{\boldsymbol{F}}$, detailed in the Supplementary.

To reinforce the semantic purity of $F_s$, we apply adversarial training to $F_i$, encouraging it to encode minimal semantic content. As a result of the residual structure, this pushes more semantic information into $F_s$. Specifically, we prepend a gradient reversal layer [41] before the projection: $F_i = \mathcal{P}_i[\text{GRL}(E_s)]$, discouraging $E_s$ from aligning with CLIP target features. The corresponding adversarial loss $\mathcal{L}_{inv}^F$ is also a BiMixCo loss with min-max game similar to Eq. (2) and omitted here.

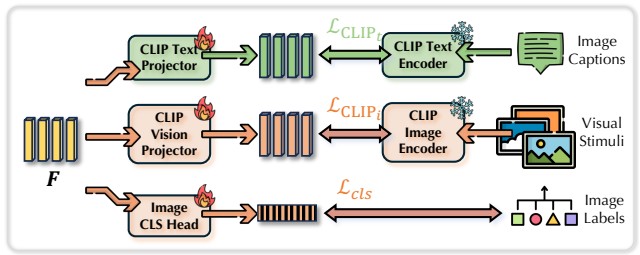

Figure 4: Representation Preservation Anchor of SSFE.

**Representation Preservation Anchor.** Similar to SIFE, to ensure semantic consistency in the latent space, we introduce a preservation anchor by aligning CLIP embeddings across three perspectives: classification, vision, and text. We employ a linear classifier $\mathcal{C}(\cdot)$ on the $E$ to predict the image label $\hat{c}$, with a cross-entropy loss $\mathcal{L}_{cls}$. For vision and text alignments, we adopt BiMixCo losses $\mathcal{L}_{\text{CLIP}_v}$ and $\mathcal{L}_{\text{CLIP}_t}$ to align to CLIP image and text embeddings. The total semantic loss is $\mathcal{L}_{\text{sem}} = \mathcal{L}_{cls} + \mathcal{L}_{\text{CLIP}_v} + \mathcal{L}_{\text{CLIP}_t}$. The overall training loss can be described as:

$$\mathcal{L} = \mathcal{L}_{rec} + \mathcal{L}_{dis}^E + \mathcal{L}_{cls}^E + \mathcal{L}_{inv}^F + \mathcal{L}_{spe}^F + \mathcal{L}_{sem} + \lambda\mathcal{L}_{prior}, \quad (5)$$

where $\lambda$ is set to 30 following previous methods [4].

## 4 Experiments

### 4.1 Experimental Setup

**Dataset.** We use the Natural Scenes Dataset (NSD) [17] for both training and evaluation. NSD contains visual image stimulus and corresponding fMRI recordings of 8 subjects, with each subject viewing 8,000-9,000 images. The original images are collected from MS-COCO dataset [42], which are consisted of complex natural images. Following [2], we use the corresponding captions of the images in COCO dataset for training. For both training and evaluation, we average three trials of fMRI signal of the same images following [5]. The final results were tested on subjects 1, 2, 5 or 7, since these subjects complete all scanning sessions, sharing the same 982 images as testing data. For each test subject, we use all other 7 subjects to train the model and tested on the unseen subject with unseen test split.

**Evaluation Metrics.** We follow the metrics of Mindeye2 [3] to evaluate both high-level and low-level consistency. On the low-level aspect, we use pixelwise correlation, Structural Similarity Index Metric (SSIM) [43], AlexNet(2), and AlexNet(5). High-level metrics are calculated by extracting features using specific networks, including EffNet-B [44], SwAV [45], Inception [46], and CLIP [47]. Please refer to the Supplementary for more details.

**Implementation Details.** All experiments were conducted for 60 epochs using 8 NVIDIA RTX H800 GPUs with a total batch size of 128 (16 samples per GPU). We adopt the AdamW optimizer [48] with a learning rate of 1e-4, following the OneCycle learning rate schedule [49]. In the inference stage, we follow MindEye2's two-stage decoding process. First, the predicted image latents are decoded into coarse images using SDXL unCLIP. These coarse outputs are then refined using base SDXL in image-to-image mode, guided by predicted captions. The refinement starts from a noised version of the coarse image, skipping the first 50% of diffusion steps.

### 4.2 Main Results

**Quantitative Results.** We evaluate ZEBRA against representative methods across various training regimes on the Natural Scenes Dataset, with results averaged over subjects 1, 2, 5, and 7. As shown in Table 1, ZEBRA achieves competitive performance without using any subject-specific data, highlighting its strong generalization ability in the *zero-shot* setting. Compared to the only other zero-shot-compatible baseline, NeuroPictor*, ZEBRA achieves substantial improvements across all metrics. On low-level similarity metrics, ZEBRA improves PixCorr from 0.057 to 0.131 and SSIM from 0.297 to 0.375. Similarly, high-level perceptual metrics show consistent gains: ZEBRA achieves

Table 1: Quantitative comparison of ZEBRA against representative methods under different training regimes. Results are averaged over subjects 1, 2, 5, and 7 from the Natural Scenes Dataset. *Fully fine-tuned* methods are trained on 40-hour data from the test subject, *few-shot* methods use only 1-hour recordings, while *zero-shot* methods do not use any data from the test subjects. "NeuroPictor†" denotes a version pretrained on 40-hour data from all subjects without fine-tuning on the test subjects. "NeuroPictor⋆" represents our implementation in a zero-shot setting (pretrained on the other 7 subjects). Previous methods—excluding NeuroPictor—are fundamentally infeasible in zero-shot scenarios because their architectures depend on subject-specific linear mappings. "Our baseline" refers to the strong baseline (§3.1) combining NeuroPictor and MindTuner.

| Method | Low-Level | | | | High-Level | | | |
|---|---|---|---|---|---|---|---|---|
| | PixCorr↑ | SSIM↑ | Alex(2)↑ | Alex(5)↑ | Incep↑ | CLIP↑ | Eff↓ | SwAV↓ |
| *Fully fine-tuned* | | | | | | | | |
| Takagi... [1] [CVPR'23] | 0.246 | 0.410 | 78.9% | 85.6% | 83.8% | 82.1% | 0.811 | 0.504 |
| Ozcelik... [2] [Sci. Rep.'23] | 0.273 | 0.365 | 94.4% | 96.6% | 91.3% | 90.9% | 0.728 | 0.422 |
| MindEye1 [3] [NeurIPS'24] | 0.319 | 0.360 | 92.8% | 96.9% | 94.6% | 93.3% | 0.648 | 0.377 |
| UMBRAE [34] [ECCV'24] | 0.283 | 0.341 | 95.5% | 97.0% | 91.7% | 93.5% | 0.700 | 0.393 |
| NeuroPictor [5] [ECCV'24] | 0.229 | 0.375 | 96.5% | 98.4% | 94.5% | 93.3% | 0.639 | 0.350 |
| NeuroPictor† [5] [ECCV'24] | 0.141 | 0.349 | 91.4% | 95.7% | 88.3% | 88.9% | 0.722 | 0.417 |
| MindBridge [31] [CVPR'24] | 0.151 | 0.263 | 87.7% | 95.5% | 92.4% | 94.7% | 0.712 | 0.418 |
| MindEye2 [4] [ICML'24] | 0.322 | 0.431 | 96.1% | 98.6% | 95.4% | 93.0% | 0.619 | 0.344 |
| MindTuner [6] [AAAI'25] | 0.322 | 0.421 | 95.8% | 98.8% | 95.6% | 93.8% | 0.612 | 0.340 |
| *Few-shot* | | | | | | | | |
| MindEye2 [4] (1 hour) | 0.195 | 0.419 | 84.2% | 90.6% | 81.2% | 79.2% | 0.810 | 0.468 |
| MindTuner [6] (1 hour) | 0.224 | 0.420 | 87.8% | 93.6% | 84.8% | 83.5% | 0.780 | 0.440 |
| *Zero-shot* | | | | | | | | |
| NeuroPictor⋆ | 0.057 | 0.297 | 71.4% | 74.7% | 62.5% | 66.0% | 0.939 | 0.607 |
| Our baseline | 0.074 | 0.316 | 70.8% | 74.0% | 63.5% | 62.5% | 0.920 | 0.602 |
| ZEBRA | 0.131 | 0.375 | 74.6% | 81.2% | 72.2% | 71.5% | 0.837 | 0.506 |

Table 2: Ablations on the key components of ZEBRA, and all results are from subject 1. 'Adv.' denotes adversarial training and 'Anchor' stands for preservation anchor.

| Base line | SIFE | | SSFE | | Low-Level | | | | High-Level | | | |
|---|---|---|---|---|---|---|---|---|---|---|---|---|
| | Adv. | Anchor | Adv. | Anchor | PixCorr↑ | SSIM↑ | Alex(2)↑ | Alex(5)↑ | Incep↑ | CLIP↑ | Eff↓ | SwAV↓ |
| ✓ | | | | | 0.089 | 0.325 | 72.5% | 74.7% | 64.7% | 63.2% | 0.891 | 0.579 |
| ✓ | ✓ | | | | 0.129 | 0.355 | 73.9% | 77.4% | 68.0% | 66.8% | 0.885 | 0.545 |
| ✓ | ✓ | ✓ | | | 0.134 | 0.368 | 74.3% | 78.3% | 70.0% | 69.3% | 0.855 | 0.525 |
| ✓ | ✓ | ✓ | ✓ | | 0.142 | 0.374 | 75.2% | 79.6% | 71.4% | 70.8% | 0.832 | 0.505 |
| ✓ | ✓ | ✓ | ✓ | ✓ | 0.153 | 0.384 | 76.1% | 81.8% | 73.4% | 72.3% | 0.814 | 0.490 |

74.6% and 81.2% on AlexNet(2) and AlexNet(5), respectively, compared to 71.4% and 74.7% for NeuroPictor⋆. On high-level semantic metrics, ZEBRA outperforms NeuroPictor⋆ with margins of +9.7% on Inception (72.2% vs. 62.5%) and +5.5% on CLIP similarity (71.5% vs. 66.0%). ZEBRA also demonstrates lower perceptual distance, with Eff decreasing from 0.939 to 0.837, and SwAV from 0.607 to 0.506, indicating stronger alignment with ground-truth representations. These results demonstrate ZEBRA's clear advantage in generalizing across subjects without requiring fine-tuning. While fully fine-tuned methods unsurprisingly perform better due to access to the data of test subjects, ZEBRA narrows this gap significantly. For instance, in the zero-shot setting, ZEBRA achieves 74.6% on AlexNet(2), compared to 78.9% by Takagi et al. [1] and 87.7% by MindBridge [31], despite using no data from the test subjects. In summary, ZEBRA sets a new state of the art in zero-shot neural decoding, outperforming prior zero-shot methods by large margins across all metrics, and approaching the performance of few-shot and even some fully fine-tuned approaches.

**Qualitative Results.** We compare ZEBRA with the zero-shot implementation of NeuroPictor⋆ and the few-shot baseline MindEye2 (1-hour) as shown in Fig. 5. Qualitatively, ZEBRA generates high-fidelity images that are visually comparable to those produced by fully supervised models, and clearly surpasses NeuroPictor⋆ in overall perceptual quality. Compared to few-shot methods, the main

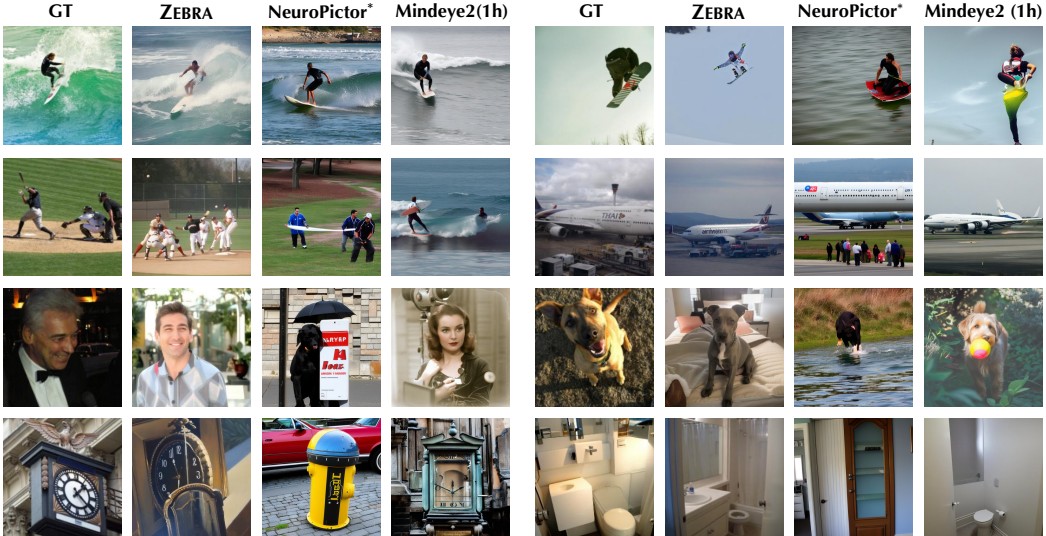

| GT | ZEBRA | NeuroPictor* | Mindeye2(1h) | GT | ZEBRA | NeuroPictor* | Mindeye2 (1h) |

Figure 5: Qualitative comparison between ZEBRA and zero-shot implementation of NeuroPictor and Mindeye2 (1h).

limitation of ZEBRA lies in semantic accuracy. As highlighted in the failure cases Fig. 7, ZEBRA tends to struggle with fine-grained semantic distinctions, especially for rare object categories.

## 4.3 Ablation Studies

We conduct ablation studies on Subject 1 (trained on Subjects 2-8) to assess the contribution of each component in ZE-BRA, including adversarial training and the proposed preservation anchor in both the Subject-Invariant Feature Extractor (SIFE) and the Subject-Specific Feature Enhancer (SSFE).

**Effectiveness of the key components.** As shown in Table 2, the base model without any regularization performs poorly (e.g., PixCorr = 0.089, Alex(5) = 72.7%). Adding adversarial training to SIFE yields clear improvements across both low- and high-level metrics (e.g., +0.040 PixCorr, +3.6% CLIP). Adding the anchor further boosts performance, especially in high-level features like CLIP. Incorporating adversarial loss in SSFE brings additional gains, especially in high-level metrics, confirming the benefit of regularizing both branches. The full model with all components achieves the best overall results, highlighting the effectiveness of our design.

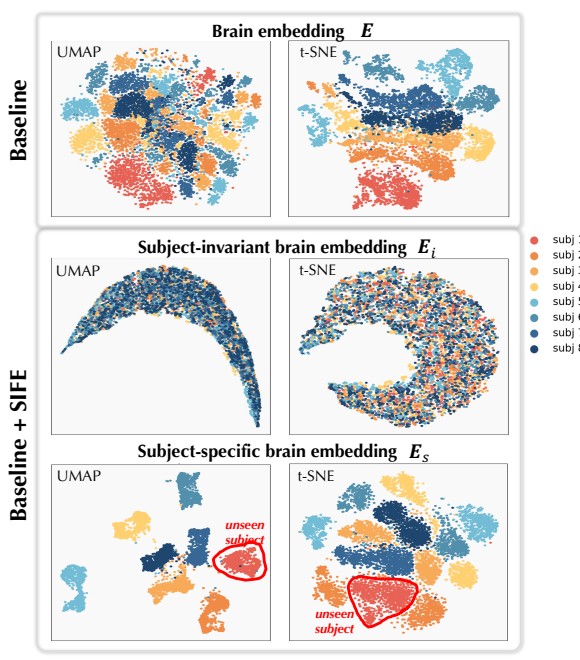

Figure 6: Visualization of subject-invariant and -specific features with UMAP and t-SNE.

**Effectiveness of the proposed subject-invariant representation learning.** In Fig. 6, we use UMAP and t-SNE to visualize the feature distributions of different subjects. The left two plots show the subject-invariant features $E_i$, while the right two show the subject-specific features $E_s$. In the case of $E_i$, data points from all subjects are highly mixed without forming subject-specific clusters, demonstrating that the learned features are invariant to subject identity. In contrast, $E_s$ exhibits clear

Table 3: Ablation on the number of training subjects for ZEBRA, evaluated on data from subject 1.

| # of | Low-Level | | | | High-Level | | | |
|---|---|---|---|---|---|---|---|---|
| Subjects | PixCorr↑ | SSIM↑ | Alex(2)↑ | Alex(5)↑ | Incep↑ | CLIP↑ | Eff↓ | SwAV↓ |
| 4 (2 to 5) | 0.109 | 0.325 | 68.2% | 73.5% | 66.1% | 63.7% | 0.871 | 0.563 |
| 5 (2 to 6) | 0.126 | 0.347 | 71.3% | 76.8% | 68.7% | 67.2% | 0.846 | 0.534 |
| 6 (2 to 7) | 0.135 | 0.363 | 74.1% | 79.0% | 70.8% | 69.6% | 0.823 | 0.508 |
| 7 (2 to 8) | **0.153** | **0.384** | **76.1%** | **81.8%** | **73.4%** | **72.3%** | **0.814** | **0.490** |

clustering by subject, indicating that the model successfully captures individual-specific information. These results confirm that our method effectively disentangles subject-invariant and subject-specific representations.

**Ablation on Number of Training Subjects.** As shown in Table 3, we evaluate the impact of training subject count on model performance. As the number of subjects increases from 4 to 7, we observe consistent performance improvements across all metrics. For instance, PixCorr improves significantly from 0.109 to 0.153, and both low- and high-level semantic scores show similar trends, such as CLIP score increasing from 63.7% to 72.3%. These results demonstrate that incorporating more diverse subject data helps the model generalize better, highlighting its scalability and robustness.

## 5   Discussion and Conclusion

**Key Contributions and Insights.**   The novelty of our work lies not in individual architectural components, but in the problem formulation, representation disentanglement design, and cross-subject transfer mechanism that together enable zero-shot fMRI-to-image decoding. First, the importance of the zero-shot setting: traditional fMRI decoding frameworks typically rely on fully supervised, subject-specific training, requiring separate model tuning for each individual under expert supervision. Such procedures are time-consuming—often exceeding 12 hours per subject—and computationally prohibitive for clinical use. In contrast, our approach introduces, for the first time, a zero-shot setting that allows the model to be directly applied to unseen subjects without retraining. This paradigm shift makes neural decoding fast, scalable, and clinically practical, achieving 73.4% decoding performance with approximately one second of inference per image. Second, neuroscience-inspired representation disentanglement: our design is motivated by neuroscientific evidence that, despite inter-individual variability, the human cortex encodes semantic information in a consistent and topographically organized manner across subjects. To preserve this universality while maintaining discriminative power, we explicitly separate subject-invariant and semantic-specific representations through the SIFE and SSFE modules, balancing fidelity and generalizability. Third, adversarial disentanglement with preservation anchors: adversarial training objectives are employed to automatically extract invariant and specific features, while a Representation Preservation Anchor ensures that essential individual information is retained during zero-shot transfer. Finally, empirical superiority in zero-shot decoding: our framework outperforms state-of-the-art baselines adapted to the zero-shot setting (e.g., MindTuner), underscoring the effectiveness of our disentanglement and transfer design in achieving robust cross-subject generalization.

**Limitations and Future Work.** Despite the encouraging results, several limitations remain. Although our reconstructed images exhibit competitive quality, especially in low-level perceptual metrics, their semantic fidelity still lags behind few-shot approaches. Improving high-level semantic accuracy remains a key challenge. Nonetheless, our work offers a promising zero-shot strategy and lays the foundation for building generalizable brain decoding models. Another limitation lies in the scope of downstream tasks. While this study focuses on image reconstruction, the proposed ZEBRA framework is inherently modality-agnostic and could be extended to more complex domains such as text or video. For instance, integrating ZEBRA with existing methods like NeuroClips [50] or NEURONS [51] could enable zero-shot fMRI-to-video generation, facilitating a richer understanding of human perceptual experiences. Moreover, the current dataset contains a limited number of subjects, which restricts the ability to fully demonstrate the generalizability of our approach. We believe that as more training subjects and fMRI data become available, the model's robustness and zero-shot performance will further improve. Expanding subject diversity is especially important for advancing toward universal brain decoders. Finally, additional fMRI recordings and broader subject coverage are essential to capture real-world visual experiences more comprehensively. Addressing these limitations

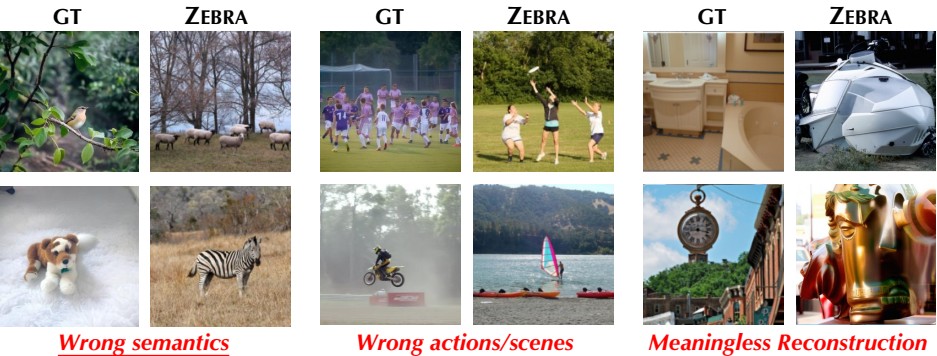

|  GT | ZEBRA | GT | ZEBRA | GT | ZEBRA |

*Wrong semantics*       *Wrong actions/scenes*       *Meaningless Reconstruction*

Figure 7: Failure cases, mainly caused by wrong semantics.

will require interdisciplinary progress across machine learning, computer vision, neuroscience, and biomedical engineering. We emphasize that alongside technical advances, it is equally important to establish ethical and regulatory frameworks to ensure the privacy and responsible use of brain data.

**Conclusion.** In this work, we introduced ZEBRA, a novel zero-shot brain visual decoding framework that addresses the critical challenge of generalizing fMRI-to-image reconstruction to unseen subjects. By disentangling subject-specific and semantic-specific components in the fMRI embedding space, ZEBRA enables accurate visual reconstruction without requiring additional data or retraining for new individuals. Our approach leverages adversarial learning and residual decomposition to isolate shared semantic representations, achieving strong generalization across subjects. Extensive experiments demonstrate that ZEBRA outperforms existing zero-shot baselines and approaches the performance of fully finetuned models, both quantitatively and qualitatively. This represents a significant step toward practical and scalable brain decoding systems with real-world applicability in neuroscience, clinical settings, and brain-computer interfaces.

## Acknowledgements

This work was partially supported by a grant from the Joint Research Scheme (JRS) under the National Natural Science Foundation of China (NSFC) and the Research Grants Council (RGC) of Hong Kong (Project No. N_HKUST654/24), a grant from the RGC of the Hong Kong Special Administrative Region, China (Project No. R6005-24), and a grant from the RGC of the Hong Kong Special Administrative Region, China (Project No. AoE/E-601/24-N).

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

# Appendix

## 1 More Details about the Baseline of ZEBRA

The Baseline employs the ViT-based fMRI encoder from fMRI-PTE [1], which is pretrained on the UK Biobank dataset [2]. This encoder transforms the fMRI scan into a shared latent space, where a diffusion prior network then converts the latent brain embeddings into vision features for image generation using Stable Diffusion.

The Baseline consists of three main components:

1. **fMRI Encoder**: This module processes the input fMRI data and transforms it into a unified 2D brain activation map [1], resulting in a single-channel image $x \in \mathbb{R}^{256 \times 256}$. The fMRI encoder then maps this image into a latent representation $\boldsymbol{E} \in \mathbb{R}^{B \times L \times C_1}$, where $B$ is the batch size, $L$ is the number of tokens, and $C_1$ is the brain feature dimension.

2. **Latent Representation Conversion**: The latent brain embedding $\boldsymbol{E}$ is subsequently transformed into a CLIP-compatible embedding $\boldsymbol{F} \in \mathbb{R}^{B \times L \times C_2}$ to provide guidance for the reconstruction process.

3. **Diffusion Prior Network**: As in *MindEye2* [3], we employ a diffusion prior [4] to map the fMRI-CLIP embedding $\boldsymbol{F}$ to a reconstructed OpenCLIP image embedding $\boldsymbol{F}_y$ corresponding to the visual stimulus.

The training of the Baseline involves three key losses:

1. **Contrastive Loss on CLIP Text Embeddings**: This loss, denoted $\mathcal{L}_{\mathrm{CLIP_t}}$, is calculated between the predicted CLIP text embedding $\boldsymbol{F}^t$ and the ground truth $\boldsymbol{F}_y^t$.

2. **Contrastive Loss on CLIP Image Embeddings**: The loss $\mathcal{L}_{\mathrm{CLIP_i}}$ is computed between the predicted CLIP image embedding $\boldsymbol{F}$ and the ground truth $\boldsymbol{F}_y^i$.

3. **Diffusion Prior Loss**: The loss $\mathcal{L}_{\mathrm{prior}}$ is used to train the diffusion prior network to minimize the reconstruction error.

Both $\mathcal{L}_{\mathrm{CLIP_t}}$ and $\mathcal{L}_{\mathrm{CLIP_i}}$ are implemented as the BiMixCo loss, which aligns fMRI signals $x$ and corresponding image embeddings $y$ using a bidirectional contrastive loss and MixCo data augmentation, as detailed below.

The MixCo procedure involves mixing two independent fMRI signals. For each fMRI signal $x$, we randomly sample another fMRI signal $x_m$ corresponding to a different index $m$. The two signals are then mixed using a linear combination:

$$x^* = \mathrm{mix}(x, x_m) = \lambda \cdot x + (1 - \lambda)x_m, \tag{1}$$

where $x^*$ represents the mixed fMRI signal and $\lambda$ is a hyperparameter sampled from a Beta distribution. The ridge regression module then maps $x^*$ to a lower-dimensional space, yielding $x^{*'}$, from which the embedding $\boldsymbol{F}$ is obtained using the MLP, i.e., $\boldsymbol{F} = \mathcal{E}(x^{*'})$.

39th Conference on Neural Information Processing Systems (NeurIPS 2025).

The BiMixCo loss function is formulated as:

$$
\begin{aligned}
\mathcal{L}_{\text{BiMixCo}} = &- \frac{1}{2L} \sum_{i=1}^{L} \lambda_i \cdot \log \frac{\exp(\text{sim}(\boldsymbol{F}_i, \boldsymbol{y}_i)/\tau)}{\sum_{k=1}^{L} \exp(\text{sim}(\boldsymbol{F}_i, \boldsymbol{y}_k)/\tau)} \\
&- \frac{1}{2L} \sum_{i=1}^{L} (1 - \lambda_i) \cdot \log \frac{\exp(\text{sim}(\boldsymbol{F}_i, \boldsymbol{y}_{m_i})/\tau)}{\sum_{k=1}^{L} \exp(\text{sim}(\boldsymbol{F}_i, \boldsymbol{y}_k)/\tau)} \\
&- \frac{1}{2L} \sum_{j=1}^{L} \lambda_j \cdot \log \frac{\exp(\text{sim}(\boldsymbol{F}_j, \boldsymbol{y}_j)/\tau)}{\sum_{k=1}^{L} \exp(\text{sim}(\boldsymbol{F}_j, \boldsymbol{y}_j)/\tau)} \\
&- \frac{1}{2L} \sum_{j=1}^{L} \sum_{\{l \mid m_l = j\}} (1 - \lambda_j) \cdot \log \frac{\exp(\text{sim}(\boldsymbol{F}_l, \boldsymbol{y}_j)/\tau)}{\sum_{k=1}^{L} \exp(\text{sim}(\boldsymbol{F}_l, \boldsymbol{y}_j)/\tau)},
\end{aligned}
\tag{2}
$$

where $\boldsymbol{F}$ represents the OpenCLIP embeddings for the image $y$.

The Diffusion Prior network is used to transform the fMRI embedding $\boldsymbol{F}$ into the reconstructed OpenCLIP image embeddings of stimulus $\boldsymbol{F}_y$. The objective is to minimize the mean squared error (MSE) between the predicted and target embeddings, formulated as:

$$
\mathcal{L}_{\text{Prior}} = \mathbb{E}_{\boldsymbol{F}_y, \boldsymbol{F}, \epsilon \sim \mathcal{N}(0,1)} \|\epsilon(\boldsymbol{F}) - \boldsymbol{F}_y\|^2.
\tag{3}
$$

## 2   More Details about Metrics

To evaluate reconstruction quality, we adopt a comprehensive set of low-level and high-level metrics.

On the low-level side, we include four metrics: pixel-wise correlation, Structural Similarity Index Measure (SSIM) [5], AlexNet(2), and AlexNet(5). Pixel-wise correlation and SSIM are computed by averaging the similarity scores between each reconstructed image and its ground-truth counterpart. AlexNet(2) and AlexNet(5) assess semantic similarity by measuring the two-way classification accuracy based on features extracted from the 2nd and 5th layers of a pre-trained AlexNet, respectively.

For high-level evaluation, we extract features using several pre-trained models. EffNet-B and SwAV metrics are calculated by averaging the feature distance between reconstructions and ground-truth images using EfficientNet-B1 [6] and SwAV-ResNet50 [7]. In contrast, the Inception [8] and CLIP [9] metrics reflect the accuracy of two-way classification using the corresponding high-level features.

## 3   Comparison with Baselines across Subjects

Table 1 presents a detailed comparison of ZEBRA with NeuroPictor⋆, and our baseline across subjects 1, 2, 5, and 7 from the Natural Scenes Dataset. The results are reported under a zero-shot setting, where NeuroPictor⋆ refers to our reimplementation pretrained on the remaining seven subjects without any subject-specific finetuning. Across both low-level (PixCorr, SSIM) and high-level metrics (AlexNet, Inception, CLIP, etc.), ZEBRA consistently outperforms the zero-shot baselines, especially on semantic metrics like Alex(5) and CLIP accuracy. Notably, its performance remains stable across subjects, indicating strong generalization ability. The final row reports the averaged scores across all four subjects, further confirming the effectiveness of ZEBRA in both perceptual quality and semantic fidelity.

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

Table 1: Results of NeuroPictor*, our baseline, and ZEBRA for each subject, compared against representative methods under different training regimes. All results are averaged over subjects 1, 2, 5, and 7 from the Natural Scenes Dataset. "NeuroPictor*" denotes our implementation in a zero-shot setting, pretrained on the other 7 subjects.

| Method | Low-Level | | | | High-Level | | | |
|---|---|---|---|---|---|---|---|---|
| | PixCorr↑ | SSIM↑ | Alex(2)↑ | Alex(5)↑ | Incep↑ | CLIP↑ | Eff↓ | SwAV↓ |
| *Subject 1* | | | | | | | | |
| NeuroPictor* | 0.069 | 0.305 | 73.1% | 75.4% | 63.6% | 66.8% | 0.910 | 0.583 |
| Our baseline | 0.089 | 0.325 | 72.5% | 74.7% | 64.7% | 63.2% | 0.891 | 0.579 |
| ZEBRA | 0.153 | 0.384 | 76.1% | 81.8% | 73.4% | 72.3% | 0.814 | 0.490 |
| *Subject 2* | | | | | | | | |
| NeuroPictor* | 0.061 | 0.303 | 73.0% | 75.2% | 62.4% | 65.9% | 0.938 | 0.590 |
| Our baseline | 0.079 | 0.323 | 72.4% | 74.5% | 63.5% | 62.3% | 0.918 | 0.586 |
| ZEBRA | 0.135 | 0.382 | 76.0% | 81.6% | 72.0% | 71.3% | 0.839 | 0.496 |
| *Subject 5* | | | | | | | | |
| NeuroPictor* | 0.049 | 0.290 | 69.7% | 74.6% | 62.7% | 66.7% | 0.922 | 0.599 |
| Our baseline | 0.063 | 0.309 | 69.1% | 73.9% | 63.8% | 63.1% | 0.903 | 0.595 |
| ZEBRA | 0.119 | 0.365 | 72.6% | 80.9% | 72.4% | 72.1% | 0.825 | 0.503 |
| *Subject 7* | | | | | | | | |
| NeuroPictor* | 0.049 | 0.290 | 69.9% | 73.7% | 61.2% | 64.7% | 0.986 | 0.655 |
| Our baseline | 0.064 | 0.309 | 69.3% | 73.0% | 62.2% | 61.2% | 0.966 | 0.650 |
| ZEBRA | 0.117 | 0.370 | 73.8% | 80.4% | 71.0% | 70.4% | 0.869 | 0.534 |
| *Average* | | | | | | | | |
| NeuroPictor* | 0.057 | 0.297 | 71.4% | 74.7% | 62.5% | 66.0% | 0.939 | 0.607 |
| Our baseline | 0.074 | 0.316 | 70.8% | 74.0% | 63.5% | 62.5% | 0.920 | 0.602 |
| ZEBRA | 0.131 | 0.375 | 74.6% | 81.2% | 72.2% | 71.5% | 0.837 | 0.506 |

[5] Z. Wang, A. C. Bovik, H. R. Sheikh, and E. P. Simoncelli, "Image quality assessment: from error visibility to structural similarity," *IEEE transactions on image processing*, vol. 13, no. 4, pp. 600–612, 2004.

[6] M. Tan and Q. Le, "Efficientnet: Rethinking model scaling for convolutional neural networks," in *International conference on machine learning*, pp. 6105–6114, PMLR, 2019.

[7] M. Caron, I. Misra, J. Mairal, P. Goyal, P. Bojanowski, and A. Joulin, "Unsupervised learning of visual features by contrasting cluster assignments," *Advances in neural information processing systems*, vol. 33, pp. 9912–9924, 2020.

[8] C. Szegedy, V. Vanhoucke, S. Ioffe, J. Shlens, and Z. Wojna, "Rethinking the inception architecture for computer vision," in *Proceedings of the IEEE conference on computer vision and pattern recognition*, pp. 2818–2826, 2016.

[9] A. Radford, J. W. Kim, C. Hallacy, A. Ramesh, G. Goh, S. Agarwal, G. Sastry, A. Askell, P. Mishkin, J. Clark, *et al.*, "Learning transferable visual models from natural language supervision," in *International conference on machine learning*, pp. 8748–8763, PMLR, 2021.

