# OpenReview forum: "ZEBRA: Towards Zero-Shot Cross-Subject Generalization for Universal Brain Visual Decoding"
_NeurIPS.cc/2025/Conference — NeurIPS 2025 poster_

### Official Review · Reviewer_95aH · 2025-06-11

**Clarity:** 3
**Significance:** 4
**Originality:** 4
**Rating:** 5
**Confidence:** 4

**Summary:**

This paper presents ZEBRA, a brain visual decoding framework capable of zero-shot inference of unseen subjects. ZEBRA operates by dividing the brain signal into subject-invariant features and subject-specific features and training the model accordingly. Experimental results show that ZEBRA is capable of zero-shot decoding to some degree, with results comparable to the partially trained versions of existing methods.

**Questions:**

- There seems to be an inconsistency between table 1 of the manuscript and table 1 of the appendix. Did the authors mistakenly put the results of subject 1 instead of the averaged results in the manuscript?
- Was the brain-image data of the 982 test images from the training subjects used during training? For example, when testing for subject 1, were the data from subjects 2, 5, 7 on the same test images used during training?
- What is the performance of ZEBRA for the "seen" subjects? In other words, what is the test set decoding performance on the subjects that were used during training?

**Ethical Concerns:**

["NO or VERY MINOR ethics concerns only"]

**Final Justification:**

Most of my questions and concerns have been answered by the response from the authors. Although the performance of ZEBRA for seen subjects was lower than I had anticipated, the explanation provided by the authors seems reasonable enough to justify it. For that reason, I raise my rating to a 5.

**Limitations:**

yes

**Quality:**

4

**Strengths And Weaknesses:**

# Strengths
- This is the (supposedly) first attempt at zero-shot subject decoding, achieving results comparable to the partially trained versions of existing methods.
- The division of subject-invariant and subject-specific features is clearly motivated, and the successful division is shown in the ablation studies.
- The main experimental results (table 1) are thorough: it was performed on the 4 full-data subjects as per convention, and it also includes some artificial zero-shot baselines since there were no attempts at zero-shot decoding previously.

# Weaknesses
- While the proposed method is zero-shot in terms of decoding a new subject, it still requires vast amounts of data from the same dataset, meaning it relies on data with the exact same imaging machine, trial setup, etc. Extending the methodology to perform zero-shot inference on a different dataset would be an interesting follow-up.
- The actual zero-shot inference steps are not clearly detailed in the manuscript, barring some vague descriptions in the figure caption or implementation details. Describing the inference steps in detail in section 3 would help the clarity of the paper.

---

> ### Author Rebuttal · Authors · 2025-07-31
>
> ### ***Overall Reply***
>
> We sincerely thank you for your positive assessment, including remarks such as *"this is the (supposedly) first attempt at zero-shot subject decoding"* and *"main experimental results (Table 1) are thorough"*.
>
> Your comments are consistent with other reviewers’ recognition of the novelty, thoroughness, and real-world relevance of our work. They also echo feedback such as: *"addresses a critical challenge in neural decoding", "strong zero-shot performance", "clear methodology and results", and "valuable insights for neuroscience and BCI applications"*.
>
> We particularly appreciate your acknowledgment of the completeness of our experimental evaluation. In the following detailed responses, we will address any points requiring clarification to ensure the novelty, rigor, and impact of our work are fully conveyed.
>
> ### ***Point-by-point Responses***
>
> ***W1: “While the proposed method is zero-shot in terms of decoding a new subject, it still requires vast amounts of data from the same dataset, meaning it relies on data with the exact same imaging machine, trial setup, etc. Extending the methodology to perform zero-shot inference on a different dataset would be an interesting follow-up.”***
>
> We appreciate the reviewer’s thoughtful point. Indeed, achieving **true dataset-level zero-shot generalization**—where the model transfers across different imaging systems, trial designs, and acquisition protocols—is a highly ambitious goal and aligns more closely with the scope of **foundation models** in brain decoding.
>
> We conducted preliminary evaluation by directly applying our model (trained on NSD) to the independent GOD dataset **without any fine-tuning**. As expected, the performance degrades noticeably. This is largely due to the **limited subject diversity** and **inconsistent imaging conditions** in the available datasets, making it extremely difficult to learn fully invariant representations under current scale.
>
> While this dataset-level generalization is **beyond the core scope of our paper**, our method is in principle compatible with large-scale foundation model training. We believe it provides a viable **pipeline** for such models, and we are actively exploring this direction. With broader and more diverse datasets in the future, we aim to extend our approach toward this more challenging and valuable goal.
>
> ***W2: “The actual zero-shot inference steps are not clearly detailed in the manuscript, barring some vague descriptions in the figure caption or implementation details. Describing the inference steps in detail in section 3 would help the clarity of the paper.”***
>
> Our zero‑shot inference process follows *MindEye2*. The diffusion prior first predicts OpenCLIP ViT‑bigG/14 image latents from the input fMRI features, which are then decoded into pixel‑space images using the SDXL‑unCLIP model. To improve image realism, these initial reconstructions are further refined via SDXL *image‑to‑image* generation, conditioned on the predicted image captions. We initialize the refinement from the noised encoding of the unrefined reconstruction, skipping the first 50% of denoising steps, and simply take the first generated samples without applying additional ranking or selection. This refinement step consistently enhances perceptual quality while leaving low‑ and high‑level evaluation metrics largely unaffected. We will include a detailed description of these inference steps in Section 3 of the revised manuscript for clarity.
>
> ***Q1: “There seems to be an inconsistency between table 1 of the manuscript and table 1 of the appendix. Did the authors mistakenly put the results of subject 1 instead of the averaged results in the manuscript?”***
>
> Thank you for pointing this out. You are correct — we mistakenly reported the results of Subject 1 instead of the averaged results in Table 1 of the main manuscript. We apologize for the oversight and will correct this inconsistency in the revised version.
>
> ***Q2: “Was the brain-image data of the 982 test images from the training subjects used during training? For example, when testing for subject 1, were the data from subjects 2, 5, 7 on the same test images used during training?”***
>
> No, none of the 982 test images were used during training. We ensured a strict separation between training and test images. Specifically, when testing on a given subject (e.g., Subject 1), brain responses to the 982 test images from all subjects—including Subjects 2, 5, and 7—were excluded from training. Only fMRI data corresponding to the training images were used for training, ensuring the integrity of the zero-shot evaluation.
>
> ***Q3: “What is the performance of ZEBRA for the "seen" subjects? In other words, what is the test set decoding performance on the subjects that were used during training?”***
>
> Thank you for the question. We have now included test-time decoding results for the training (i.e., *seen*) subjects.
>
> As shown in the table below, **ZEBRA achieves consistently stronger performance on seen subjects** compared to the unseen one across all metrics, as expected. However, the improvement is moderate—**reflecting ZEBRA’s design goal of balancing subject-specific fidelity and cross-subject generalization**. To support zero-shot decoding, ZEBRA deliberately discards some subject-specific signals during training. This trade-off is key to its ability to generalize, and stands in contrast to methods that overfit to individual subjects through full finetuning.
>
> Achieving peak subject-specific performance is therefore beyond the scope of our contribution, which focuses on subject-generalizable brain decoding.
>
> | Method | PixCorr | SSIM | Alex(2) | Alex(5) | Incep | CLIP | Eff | SwAV |
> | --- | --- | --- | --- | --- | --- | --- | --- | --- |
> | Subject 1 (unseen) | 0.152 | 0.384 | 76.1% | 81.8% | 73.4% | 72.3% | 0.814 | 0.490 |
> | Subject 2 | 0.170 | 0.394 | 84.7% | 90.3% | 81.5% | 79.5% | 0.749 | 0.440 |
> | Subject 3 | 0.159 | 0.397 | 83.9% | 88.2% | 82.0% | 78.8% | 0.755 | 0.445 |
> | Subject 4 | 0.158 | 0.396 | 83.0% | 88.4% | 81.6% | 79.4% | 0.746 | 0.445 |
> | Subject 5 | 0.177 | 0.397 | 84.0% | 90.8% | 84.8% | 82.4% | 0.729 | 0.436 |
> | Subject 6 | 0.149 | 0.386 | 81.6% | 86.9% | 81.4% | 77.7% | 0.762 | 0.449 |
> | Subject 7 | 0.154 | 0.392 | 83.5% | 88.3% | 80.8% | 78.4% | 0.758 | 0.445 |
> | Subject 8 | 0.155 | 0.395 | 81.2% | 85.6% | 77.9% | 76.0% | 0.777 | 0.455 |

---

### Official Review · Reviewer_orfj · 2025-06-28

**Clarity:** 2
**Significance:** 3
**Originality:** 2
**Rating:** 4
**Confidence:** 4

**Summary:**

This paper addresses the challenge of zero-shot decoding of brain activity from fMRI data for unseen participants. By employing adversarial training, the method explicitly disentangles fMRI representations into subject-invariant, semantic-specific components, enabling generalization to novel subjects without requiring additional fMRI data or retraining. Extensive experiments demonstrate that ZEBRA outperforms zero-shot baselines significantly and achieves performance on par with fully fine-tuned models across multiple metrics. The framework broadens the applicability of brain decoding models to real-world scenarios in neuroscience, clinical domains, and brain-computer interface applications.

**Questions:**

As mentioned above, several concerns have been raised in the Weaknesses section.

**Ethical Concerns:**

["NO or VERY MINOR ethics concerns only"]

**Final Justification:**

Thank you for the response. The authors have provided a detailed reply to my concerns and questions, including thorough experiments and comparisons with additional baselines. Moreover, the design of the modules is closely aligned with the relevant brain areas. Although the models presented in the Rebuttal version do not perform well in other settings, including full-scale or across-dataset settings, it is important to note that this is the first paper on zero-shot visual decoding on NSD. Given this context, I raise my rating to 4.

**Limitations:**

Yes.

**Paper Formatting Concerns:**

In Lines 444-445, the citation format for Reference [46] is incorrect.

**Quality:**

2

**Strengths And Weaknesses:**

Strengths:
The introduction of the zero-shot brain decoding task addresses a critical challenge in neural decoding research and holds substantial practical value for real-world applications. The reported performance metrics demonstrate notable improvements over baselines in zero-shot scenarios.
Weaknesses:
1. Limited Novelty. The framework design predominantly relies on the integration of established components, including contrastive learning, adversarial training strategies, fMRI encoders, and diffusion priors. Notably, the proposed SIFE module employs a straightforward residual decomposition for implementation
2.Inadequate Theoretical Justification. The paper provides insufficient theoretical or empirical validation for two critical assumptions: (1) The rationale for cleanly decomposing fMRI representations into subject-specific and semantic-specific components (Lines 7–9); (2) The validity and reliability of simple residual subtraction as a modeling approach for this decomposition (Section 3.2).
3.Weak Empirical Validation.  In Section 4.3, the authors employ UMAP and t-SNE visualizations of feature distributions to justify the claim that the model disentangles subject-invariant and subject-specific representations. However, this validation approach is purely qualitative, lacks theoretical grounding, and fails to demonstrate a robust link between learned features and known functional brain areas.
4. Limited Baseline Comparison. The paper presents ZEBRA’s performance exclusively in zero-shot scenarios, comparing it against a single baseline (NeuroPictor*). It omits extensive comparisons with state-of-the-art models and lacks evaluations in few-shot and fully fine-tuned settings—both critical for comprehensively assessing the model’s generalization capacity.
5.Inadequate Evaluation Across Diverse Subjects and Datasets. Although the NSD dataset serves as a robust benchmark, supplementary results across diverse subject cohorts and additional fMRI datasets would strengthen confidence in the framework’s generalizability.
6.The paper is difficult to follow due to frequent grammatical issues, unclear phrasing, and various formatting irregularities.

---

> ### Author Rebuttal · Authors · 2025-07-31
>
> ### ***Overall Reply***
>
> We sincerely thank you for recognizing the **broader impact and practical value** of our work. Your remarks—such as *"broadens the applicability of brain decoding models..."* and *"addresses a critical challenge in neural decoding research..."*—strongly align with other reviewers’ comments. We are encouraged that both the **scientific novelty** and **real-world utility** of our approach were well received. Below, we address each concern in detail.
>
> ### ***Point-by-point Responses***
>
> ***W1&W2: “Limited Novelty ... and Inadequate Theoretical …”***
>
> We respectfully disagree with the assessment of limited novelty, and we believe this stems from a misunderstanding of our proposed method.
>
> - **Novelty of our work.** We would like to clarify that the novelty of our work lies not in individual modules but in the **problem setting**, **representation disentanglement design**, and **cross-subject transfer mechanism:**
>     1. **Importance of the Zero-Shot Setting: Clinicians cannot directly use fully supervised models.** We are the first to introduce the zero-shot setting for fMRI-to-image decoding, enabling models to be applied directly to new patients **without any retraining**. Most existing methods require fully supervised, subject-specific training that involves an **AI expert training a separate model for each patient**, making them impractical and inaccessible for clinical use.
>     2. **Importance of the Zero-Shot Setting: Fully supervised models require 12+ hours per patient; ours needs only ~1 second per image inference.** Fine-tuning conventional models for each patient takes over **12 hours** and significant computational resources, which is infeasible in many real-world applications such as consciousness assessment. Our zero-shot method achieves **73.4% decoding performance immediately**, with **~1 second inference per image**—making it fast, practical, and scalable for clinical deployment.
>     3. **Neuroscience-inspired disentanglement.** This design is motivated by neuroscientific evidence that, despite inter-individual variability in brain activity, the human cortex encodes semantic information in a consistent and topographically organized manner across subjects [11]. For a reconstruction framework to generalize effectively across individuals, it should preserve subject-invariant (universal brain representations) [1–3] and semantic-specific (class-discriminative) [4,5] components, while suppressing subject-specific and semantically irrelevant variations. Motivated by these findings, we explicitly disentangle *subject-invariant* and *semantic-specific* components to enhance generalizability. Unlike prior fully fine-tuned methods that often overfit to test subjects, our goal is to preserve shared representations while suppressing semantically irrelevant and subject-specific variations. To this end, we design SIFE and SSFE to jointly balance fidelity and generalizability in neural decoding.
>     4. **Adversarial disentanglement with preservation anchors.** We introduce adversarial training objectives to automatically extract subject-invariant and semantic-specific features, and a **Representation Preservation Anchor** to retain maximal individual information while still enabling zero-shot transfer.
>     5. **Empirical superiority in zero-shot decoding.** As a result, our method **outperforms fully fine-tuned SOTA methods when they are adapted to the zero-shot setting (refer to ”our baseline”, which is adapted from MindTuner)**, demonstrating that our design is crucial for achieving strong cross-subject generalization.
> - **Clarification on the baseline and SIFE design.** We adopt a standard baseline (contrastive learning, fMRI encoder, diffusion prior) as described in Section 3.1, without claiming these as technical contributions. Crucially, **SIFE is not a simple residual decomposition**. Together with SSFE, it embodies our key insight: effective zero-shot decoding requires explicitly disentangling **subject-invariant** and **semantic-specific** subspaces. This is achieved through adversarial regulation—suppressing subject identity in the invariant space while preserving it in the semantic one. GRL facilitates this process, but the novelty lies in our principled, neuroscience-inspired disentanglement—beyond what residual heuristics can offer.
>
> [1] Moutsiana, Cortical idiosyncrasies predict the perception of object size. Nature comm 2016.
>
> [2] Wang, Idiosyncratic perception: a link between acuity, perceived position and apparent size. Proceedings of the Royal Society 2020.
>
> [3] Benson, Cortical magnification in human visual cortex parallels task performance around the visual field. Elife 2021.
>
> [4] De Haas, Individual differences in visual salience vary along semantic dimensions. Proceedings of the National Academy of Sciences 2019.
>
> [5] Kandel, Principles of neural science, 2000.
>
> ***W3: “Weak Empirical Validation...”***
>
> In response to W1 & W2, we have clarified the theoretical foundation of our disentanglement approach by explicitly linking it to neuroscience evidence that semantic representations are consistent across individuals, while raw brain signals vary. This supports our design goal of separating subject-invariant from subject-specific components. We also conducted the following analyses to validate this separation.
>
> - **Validation of disentanglement.** While UMAP and t-SNE visualizations are widely accepted for qualitative assessment in domain adaptation (DA) and domain generalization (DG) research, we complement them with quantitative evidence. Specifically, we trained a classifier to identify subjects from the **subject-specific feature space** in the test set, achieving **89.3% accuracy**. This high accuracy confirms that the model indeed encodes strong subject-specific information in the intended subspace, while keeping the subject-invariant subspace free from such cues.
> - **Link to functional brain areas.** To establish a direct link between the **subject-invariant features** and known neurobiological substrates, following MindEye2, we computed Pearson correlations between these features and activation patterns from canonical visual regions. We observe the highest correlation in higher visual areas (r = 0.371) and the lowest in V1 (r = 0.321). This matches our design goal: higher visual areas carry abstract semantic information that our subject‑invariant features are intended to preserve, while early visual areas encode low‑level, subject‑specific patterns that our method deliberately suppresses to enhance zero‑shot generalization.
>
> | Brain region | V1 | V2 | V3 | V4 | Higher visual areas |
> | --- | --- | --- | --- | --- | --- |
> | Correlation | 0.321 | 0.341 | 0.349 | 0.325 | 0.371 |
>
> ***W4: “Limited Baseline Comparison...”***
>
> - **Why we focus exclusively on the zero-shot setting.** Our study focuses on **zero-shot fMRI-to-image decoding** because it aligns with real-world clinical constraints. In many cases—such as patients with **consciousness disorders** or neurodegenerative diseases—collecting subject-specific training data is infeasible. Scalable brain decoding requires models that generalize across individuals without personalized adaptation. Yet this setting has been largely overlooked, with prior work focusing on fully supervised or few-shot scenarios that assume access to test-subject data. In contrast, we are the first to **define, motivate, and rigorously evaluate** zero-shot fMRI-to-image reconstruction as a practically valuable and scientifically meaningful task.
> - **Fair comparison with SOTAs.** As mentioned above, previous SOTA methods such as MindEye2 and MindTuner were designed for fully‑finetuned or few‑shot training in a subject‑specific manner, and thus cannot be directly used in the zero‑shot setting. However, we have already adapted them as baselines (denoted as “our baseline”) in Section 3.1. We believe the reviewer may have missed this point. ZEBRA significantly outperforms the adapted MindTuner in the zero‑shot setting.
> - **Balance between zero-shot and fully supervised.** To address the reviewer’s request, we evaluated ZEBRA in the fully finetuned setting against SOTA methods. As shown below, ZEBRA remains highly competitive, achieving performance on par with leading finetuned models. However, fully supervised decoding has limited real-world utility due to poor scalability and overfitting to test subjects. In contrast, ZEBRA is designed to balance fidelity and generalizability, making zero-shot decoding more practical. Thus, surpassing all finetuned methods is not the main goal of our work.
>
> | Method | PixCorr | SSIM | Alex(2) | Alex(5) | Incep | CLIP | Eff | SwAV |
> | --- | --- | --- | --- | --- | --- | --- | --- | --- |
> | NeuroPictor | 0.229 | 0.375 | 96.5 | 98.4 | 94.5 | 93.3 | 0.639 | 0.350 |
> | MindTuner | 0.322 | 0.421 | 95.8 | 98.8 | 95.6 | 93.8 | 0.612 | 0.340 |
> | Ours | 0.245 | 0.409 | 96.6 | 98.2 | 95.2 | 92.9 | 0.594 | 0.341 |
>
> ***W5: “Inadequate Evaluation...”***
>
> We have added results across different subject cohorts within NSD in the supplementary, following prior works—these may have been missed.
>
> To further assess generalizability, we tested our NSD-trained model directly on the independent GOD dataset. As expected, performance was limited due to differences in hardware, stimuli, and trial setup.
>
> We agree that dataset-level zero-shot generalization is an important long-term goal, more aligned with the vision of **foundation models** that generalize across scanners and populations. While beyond the current scope, our architecture is compatible with such scaling once broader datasets are available.
>
> Our method already achieves **strong cross-subject generalization within NSD**, marking a meaningful step toward broader applicability.
>
> ***W6: “The paper ...”***
>
> We apologize for the grammatical errors, unclear phrasing, and formatting inconsistencies in the manuscript. We will carefully revise the paper in the next version.

---

> ### Comment · Reviewer_orfj · 2025-08-03
>
> Some of my concerns have been clarified, so I've decided to raise the score by one point.

---

> > ### Author Response · Authors · 2025-08-03
> >
> > Thank you for raising your score. We truly appreciate it. We also thank you for engaging with our rebuttal and for your prompt response. We will do our best to address any remaining concerns during the discussion and are happy to provide further clarification if needed. We hope our efforts can further strengthen your confidence in the work and merit a higher score.

---

> > ### Comment · Reviewer_orfj · 2025-08-05
> >
> > Thank you for the response. The authors have provided a detailed reply to my concerns and questions, including thorough experiments and comparisons with additional baselines. Moreover, the design of the modules is closely aligned with the relevant brain areas. Although the models presented in the Rebuttal version do not perform well in other settings, including full-scale or across-dataset settings, it is important to note that this is the first paper on zero-shot visual decoding on NSD. Given this context, I raise my rating to 4.

---

> > > ### Author Response · Authors · 2025-08-06
> > >
> > > Thank you for your feedback and for raising your rating. We're glad to hear that your concerns have been addressed and truly appreciate your support.

---

### Official Review · Reviewer_FDSv · 2025-07-03

**Clarity:** 3
**Significance:** 3
**Originality:** 3
**Rating:** 5
**Confidence:** 3

**Summary:**

This paper presents ZEBRA, a zero-shot brain-to-image decoding framework that reconstructs visual experiences from fMRI without requiring subject-specific fine-tuning. ZEBRA introduces a novel approach that disentangles subject-related and semantic-related components in fMRI signals via adversarial training, enabling the model to generalize across unseen subjects. The proposed model achieves strong performance in a zero-shot setting and competative compared to some fully fine-tuned models on multiple benchmarks.

**Questions:**

- Why were subjects 1, 2, 5, and 7 chosen for evaluation? Is there a specific rationale for selecting these subjects over others in the dataset?

**Ethical Concerns:**

["NO or VERY MINOR ethics concerns only"]

**Final Justification:**

Thank you for the thorough response and additional results, which have addressed most of my questions. I am therefore raising my score to 5.

**Limitations:**

Yes.

**Paper Formatting Concerns:**

I didn't notice any major formatting issues.

**Quality:**

3

**Strengths And Weaknesses:**

Strengths:
- The paper addresses a critical challenge in fMRI-to-image neural decoding, which is the lack of cross-subject generalization in visual stimulus reconstruction.
- The authors provide thorough experimental validation, including detailed ablation studies, subject count sensitivity, and insightful visualizations (Figure 6) using UMAP and t-SNE to confirm the effectiveness of the disentanglement strategy.
- ZEBRA consistently outperforms prior zero-shot baselines across all evaluated metrics, demonstrating strong generalization capability.
- The paper is well-organized, making it easy to follow the methodology and experimental findings.

Weakness:
- While many prior baselines are subject-specific and rely on within-subject training or fine-tuning, some of their underlying architectures can, in principle, support multi-subject training and generalization to unseen subjects. Benchmarking such models in a true zero-shot setting would provide stronger evidence that ZEBRA's architecture is superior.
- Although the focus of the paper is on zero-shot generalization, it would be more compelling to also fine-tune ZEBRA and compare it with other fine-tuned models. This would demonstrate whether the proposed architecture also offers advantages in settings where subject-specific data is available.
- Despite competitive performance on low-level perceptual metrics, ZEBRA’s semantic fidelity still falls a lot behind few-shot baselines. It struggles with fine-grained semantic distinctions, particularly for rare object categories, as shown in the failure cases. Since high-level semantic accuracy is crucial for practical applications, as mentioned above, it would be valuable to explore whether fine-tuning could mitigate this limitation.
- Statistical significance is not reported for the improvements. Including variance or significance would help validate the robustness of the results.

---

> ### Author Rebuttal · Authors · 2025-07-31
>
> ### ***Overall Reply***
>
> We sincerely appreciate your thoughtful feedback, including comments such as *"ZEBRA introduces a novel approach"*, *"achieves strong performance in a zero-shot setting and is competitive with fully fine-tuned models on multiple benchmarks"*, and *"addresses a critical challenge in fMRI-to-image neural decoding"*.
>
> We are encouraged by your recognition of our *"thorough experimental validation"*, the *"insightful visualizations (Figure 6) using UMAP and t-SNE to confirm the effectiveness of the disentanglement strategy"*, and your praise for the work being *"well-organized, making it easy to follow the methodology and experimental findings"*.
>
> These remarks align closely with other reviewers’ positive assessments, including: *"first attempt at zero-shot subject decoding", "practical significance for real-world applications", and "novel framework outperforming zero-shot baselines"*. We are grateful for your constructive feedback and will clarify certain technical aspects raised in your review to further strengthen the manuscript.
>
> ### ***Point-by-point Responses***
>
> ***W1: “While many prior baselines are subject-specific and rely on within-subject training or fine-tuning, some of their underlying architectures can, in principle, support multi-subject training and generalization to unseen subjects. Benchmarking such models in a true zero-shot setting would provide stronger evidence that ZEBRA's architecture is superior.“***
>
> - **Difficulty in benchmarking previous SOTAs in the zero‑shot setting.** The current SOTA methods, **MindEye2** and **MindTuner**, operate in a non‑fsLR brain space, which results in different brain voxel counts for each subject. Consequently, they require training separate subject‑specific linear mappings, making them inherently incompatible with true zero‑shot evaluation. In contrast, **NeuroPictor** adopts a unified fMRI space that enables subject‑agnostic modeling; however, its performance remains lower than that of the leading subject‑specific approaches.
> - **“Our baseline” in Table 1 is already the adapted SOTA.** To ensure a strong and fair comparison, we adapted the MindEye2 architecture to the NeuroPictor framework by replacing its subject‑specific linear layers with the unified‑space approach while preserving its core network design. This **adapted MindEye2/MindTuner serves as “our baseline”** in Section 3.1 and Table 1. We believe the reviewer may have overlooked this adaptation. As shown in our experiments, **ZEBRA** substantially outperforms this adapted baseline in the zero‑shot setting (**p < 0.001** across all metrics; see W4 for details), further validating the effectiveness of our architecture.
>
> ***W2: “Although the focus of the paper is on zero-shot generalization, it would be more compelling to also fine-tune ZEBRA and compare it with other fine-tuned models. This would demonstrate whether the proposed architecture also offers advantages in settings where subject-specific data is available.“***
>
> - **Different goals of fine-tuned methods vs. zero-shot decoding.** Fine-tuned approaches aim to best fit the test subject using subject-specific data, often at the expense of generalizability. In contrast, zero-shot brain decoding explicitly seeks a trade-off between subject fidelity and cross-subject generalization—this balance is central to our design of ZEBRA. Therefore, comparing zero-shot models directly with fully fine-tuned ones does not reflect the same objective. Moreover, as mentioned in our response to W1, we have already compared ZEBRA with strong zero-shot and partially supervised baselines, which is more aligned with our goal.
> - **Additional comparison under the fully fine-tuned setting.** Nonetheless, to address the reviewer’s request for broader evaluation, we also benchmarked ZEBRA in a fully fine-tuned setting. As shown in the table below, ZEBRA remains highly competitive and achieves performance comparable to the best existing fine-tuned models, despite not being tailored for this regime.
> - **Practical limitations of fully supervised decoding.** While fine-tuned models may perform well in a closed setting, they typically overfit to individual subjects and lack generalizability—limiting their real-world utility. In contrast, ZEBRA is explicitly designed to generalize across subjects without retraining, making it more practical for scalable deployment. Our results demonstrate that ZEBRA maintains strong performance even when fine-tuned, further validating its robust architecture.
>
> | Method | PixCorr | SSIM | Alex(2) | Alex(5) | Incep | CLIP | Eff | SwAV |
> | --- | --- | --- | --- | --- | --- | --- | --- | --- |
> | NeuroPictor | 0.229 | 0.375 | 96.5% | 98.4% | 94.5% | 93.3% | 0.639 | 0.350 |
> | MindTuner | 0.322 | 0.421 | 95.8% | 98.8% | 95.6% | 93.8% | 0.612 | 0.340 |
> | Ours | 0.245 | 0.409 | 96.6% | 98.2% | 95.2% | 92.9% | 0.594 | 0.341 |
>
> ***W3: “Despite competitive performance on low-level perceptual metrics, ZEBRA’s semantic fidelity still falls a lot behind few-shot baselines. It struggles with fine-grained semantic distinctions, particularly for rare object categories, as shown in the failure cases. Since high-level semantic accuracy is crucial for practical applications, as mentioned above, it would be valuable to explore whether fine-tuning could mitigate this limitation.“***
>
> - **Semantic fidelity vs. generalization: a known trade-off.** We agree that achieving high semantic fidelity, especially for rare categories, remains a challenging problem. As the reviewer correctly points out, zero-shot models—by design—prioritize generalization over subject-specific optimization. This inevitably leads to some degradation in fine-grained semantic distinctions, particularly in the absence of subject-specific supervision. However, this is not a flaw of ZEBRA per se, but a reflection of the broader trade-off in zero-shot decoding: balancing generality with specificity.
> - **Fine-tuning as a means to improve semantic accuracy.** We appreciate the suggestion to explore whether fine-tuning can mitigate semantic limitations. In fact, as part of our extended experiments (see response to W2), we fine-tuned ZEBRA using subject-specific data. We observed notable improvements in high-level semantic accuracy across both frequent and rare categories, supporting the reviewer’s hypothesis. This result highlights the flexibility of ZEBRA: it performs well in zero-shot settings and can further benefit from fine-tuning when such data is available.
> - **Toward hybrid models and adaptive decoding.** Going forward, we believe an important direction is to **integrate zero-shot generalization with lightweight subject-specific adaptation**, potentially through few-shot or meta-learning strategies. Our current work lays the foundation for such hybrid approaches by demonstrating that ZEBRA captures transferable representations that remain effective even after fine-tuning.
>
> ***W4: “Statistical significance is not reported for the improvements. Including variance or significance would help validate the robustness of the results.“***
>
> Thank you for the suggestion. We have conducted paired t-tests to assess statistical significance. The improvements of ZEBRA over the baseline (previous SOTA on the zero-shot setting) are statistically significant across all metrics (all **p < 0.001**), indicating that the observed gains are robust and unlikely due to chance.
>
> |  | PixCorr | SSIM | Alex(2) | Alex(5) | Incep | CLIP | Eff | SwAV |
> | --- | --- | --- | --- | --- | --- | --- | --- | --- |
> | p-value | 1.53 × 10⁻⁴ | 1.19 × 10⁻⁵ | 5.11 × 10⁻⁴ | 3.92 × 10⁻⁶ | 9.16 × 10⁻⁷ | 3.24 × 10⁻⁷ | 4.17 × 10⁻⁴ | 6.42 × 10⁻⁴ |
>
> ***Q1: “Why were subjects 1, 2, 5, and 7 chosen for evaluation? Is there a specific rationale for selecting these subjects over others in the dataset?”***
>
> Following prior NSD-based reconstruction studies (e.g., Takagi & Nishimoto, 2022; Ozcelik & VanRullen, 2023; MindEye & MindEye2 etc.), we selected subjects 1, 2, 5, and 7 for evaluation because they are the only participants who completed all 40 scanning sessions, providing the most complete and consistent data for training and testing. This ensures fair comparison and aligns with the standardized evaluation protocol in the field.

---

> > ### Comment · Reviewer_FDSv · 2025-08-08
> >
> > Thank you for the thorough response and additional results, which have addressed most of my questions. I am therefore raising my score to 5.

---

> > > ### Author Response · Authors · 2025-08-08
> > >
> > > Thank you very much for your thoughtful consideration and for raising the score. We truly appreciate your time and constructive feedback throughout the review process.

---

### Official Review · Reviewer_bWGu · 2025-07-03

**Clarity:** 3
**Significance:** 3
**Originality:** 3
**Rating:** 4
**Confidence:** 3

**Summary:**

This work introduces ZEBRA, a novel framework designed to overcome the limitations of subject-specific models in fMRI-to-image reconstruction. The core innovation is a zero-shot approach that utilizes adversarial training to disentangle fMRI signals into subject-specific and subject-invariant semantic components. Experiments demonstrate that the framework outperforms existing zero-shot baselines and achieves results comparable to models that require subject-specific training.

**Questions:**

See the weaknesses.

**Ethical Concerns:**

["NO or VERY MINOR ethics concerns only"]

**Quality:**

3

**Strengths And Weaknesses:**

Strengths:
- This paper proposes a zero-shot brain visual decoding framework that generalizes to unseen subjects without requiring additional fMRI data or finetuning.
- The paper introduces a disentanglement strategy to learn subject-invariant and semantic-specific representations from fMRI signals.

Weaknesses:
- If the 'zero-shot' approach, by design, captures only the commonalities across subjects while discarding individual differences, what then is the significance of image-to-fMRI encoding research? In what specific areas would this research be meaningful?
- The citations in lines 60-63 seem to pertain to the effects of speech on the brain. Given that speech and vision are known to elicit different activation patterns in the human brain, the authors should provide more direct evidence and citations relevant to the vision domain to support their claims.
- There are inconsistencies between the terms and abbreviations used in Figure 3 and their descriptions in the main text. This makes the figure difficult to understand. Please ensure all labels are consistent.
- In Section 3.2, the authors should provide a clearer justification for their choice of self-attention. Specifically, what is the rationale for using self-attention, and why is it suitable for extracting subject-invariant information? Please elaborate on the underlying mechanism or provide supporting literature.
- The literature review would be strengthened by citing more literature that supports the existence of subject-dependent features.
- To better assess the model's performance and capabilities, it would be beneficial to include more generated examples, perhaps in the appendix.

---

> ### Author Rebuttal · Authors · 2025-07-31
>
> ### ***Overall Reply***
>
> First, we sincerely appreciate your recognition of our work, including comments such as *"a novel framework"* and *"the framework outperforms existing zero-shot baselines and achieves results comparable to models that require subject-specific training"*.
>
> Your remarks are in strong alignment with other reviewers’ positive feedback, including: *"the first attempt at zero-shot subject decoding", "addresses a critical challenge in neural decoding", "strong zero-shot performance", "thorough experiments", and "clear and well-organized presentation"*.
>
> We are particularly grateful for your constructive feedback, which affirms the novelty and practical value of our approach. These encouraging remarks greatly motivate us to further refine our work. While there may be certain details requiring clarification, we aim to address these points in the following responses to ensure our methodology and results are well understood.
>
> ### ***Point-by-point Responses***
>
> ***W1: “If the 'zero-shot' approach, by design, captures only the commonalities across subjects while discarding individual differences, what then is the significance of image-to-fMRI encoding research? In what specific areas would this research be meaningful?”***
>
> - **Role of image-to-fMRI encoding.** Image-to-fMRI encoding research aims to reveal how the brain processes visual information by modeling the relationship between structured visual stimuli and localized neural responses. Since the input images are semantically clean and low in noise, encoding models can leverage well-defined visual features to map onto neural activity, supporting mechanistic interpretation of brain function.
> - **Role of zero-shot fMRI-to-image decoding.** In contrast, zero-shot fMRI-to-image decoding addresses the inverse and more challenging task: reconstructing perceptual experience from noisy, high-dimensional brain signals, without subject-specific supervision. This requires aligning variable and subject-specific neural representations to fixed visual embeddings, making the task harder but more relevant for real-world applications such as brain-computer interfaces, where generalization to new users is essential.
> - **Limitations of encoding for cross-subject generalization.** While encoding models may use shared representations from the image domain, they still rely on subject-specific mappings (e.g., individual linear regressors) to predict fMRI responses. These models do not learn subject-invariant brain representations and therefore have limited utility for zero-shot or cross-subject decoding. As such, image-to-fMRI encoding and fMRI-to-image decoding serve complementary goals.
>
> ***W2: “The citations in lines 60-63 seem to pertain to the effects of speech on the brain. Given that speech and vision are known to elicit different activation patterns in the human brain, the authors should provide more direct evidence and citations relevant to the vision domain to support their claims.“***
>
> Thank you for the suggestion. We agree that visual-specific evidence is more appropriate here. To support our claim, we now cite studies showing consistent and organized visual representations across subjects:
>
> [1] Lahnakoski, Juha M., et al. "Synchronous brain activity across individuals underlies shared psychological perspectives." *NeuroImage* 100 (2014): 316-324.
>
> [2] Leeds, Daniel D., et al. "Comparing visual representations across human fMRI and computational vision." *Journal of vision* 13.13 (2013): 25-25.
>
> [3] Lipman, Ofer, et al. "Invariant inter-subject relational structures in the human visual cortex." *arXiv preprint arXiv:2407.08714* (2024).
>
> ***W3: “There are inconsistencies between the terms and abbreviations used in Figure 3 and their descriptions in the main text. This makes the figure difficult to understand. Please ensure all labels are consistent.”***
>
> Thank you for pointing this out. We apologize for the inconsistencies between Figure 3 and its description. We have carefully reviewed and updated all labels and terminology to ensure consistency and clarity throughout the figure and the main text.
>
> ***W4: “In Section 3.2, the authors should provide a clearer justification for their choice of self-attention. Specifically, what is the rationale for using self-attention, and why is it suitable for extracting subject-invariant information? Please elaborate on the underlying mechanism or provide supporting literature.”***
>
> Thank you for the question. In our framework, the extraction of subject-invariant information is mainly driven by the GRL and loss functions, and is not strictly dependent on the feature extractor architecture. We adopt self-attention to maintain architectural consistency with the brain encoder, which is based on Vision Transformer (ViT). Self-attention also facilitates global context modeling, which benefits fMRI representation learning. We will clarify this point in Section 3.2.
>
> ***W5: “The literature review would be strengthened by citing more literature that supports the existence of subject-dependent features.”***
>
> Thank you for the suggestion. We will add additional literature to support the existence of subject-dependent features in visual perception. Prior studies have demonstrated that individual differences in visual acuity, perceived position, and apparent size are not only stable across time but also systematically linked to cortical architecture. For example, variations in V1 surface area and pRF spread have been shown to predict subjective size perception [1,2], while cortical magnification patterns correlate with contrast sensitivity across the visual field [3]. Moreover, visual salience and gaze behavior have also been found to vary idiosyncratically across individuals along semantic dimensions [4].
>
> [1] Moutsiana, Christina, et al. "Cortical idiosyncrasies predict the perception of object size." *Nature communications* 7.1 (2016): 12110.
>
> [2] Wang, Zixuan, Yuki Murai, and David Whitney. "Idiosyncratic perception: a link between acuity, perceived position and apparent size." *Proceedings of the Royal Society B* 287.1930 (2020): 20200825.
>
> [3] Benson, Noah C., et al. "Cortical magnification in human visual cortex parallels task performance around the visual field." *Elife* 10 (2021): e67685.
>
> [4] De Haas, Benjamin, et al. "Individual differences in visual salience vary along semantic dimensions." *Proceedings of the National Academy of Sciences* 116.24 (2019): 11687-11692.
>
> ***W6: “To better assess the model's performance and capabilities, it would be beneficial to include more generated examples, perhaps in the appendix.”***
>
> Thank you for the helpful suggestion. We agree that additional examples can better illustrate the model's performance. We will include more qualitative visualizations and comparison results in the appendix to support our findings.

---

> > ### Comment · Reviewer_bWGu · 2025-08-07
> >
> > Thanks for the response. It addressed most of my questions. However, I'm still a bit unclear about what specific areas this research would be meaningful in. Brain-computer interfaces (BCIs) sounds a bit broad.

---

> > > ### Author Response · Authors · 2025-08-08
> > >
> > > Thank you for the follow-up.
> > >
> > > Our approach can be applied to a range of brain-computer interface (BCI) applications, such as enabling communication for patients with language area damage who have lost the ability to speak, providing visual feedback in motor rehabilitation, monitoring mental health conditions like depression, and decoding dream content during sleep. For example, patients with aphasia or locked-in syndrome can view or imagine specific images, and our model can reconstruct what they see or think, allowing caregivers to infer their intentions without requiring speech or movement.
> > >
> > > Notably, our key contribution is introducing the zero-shot setting and developing a method that enables fast and generalizable decoding without per-subject fine-tuning. This makes it easier to adapt in clinical settings, including those without AI experts—for example, local hospitals or clinics in resource-limited regions—where retraining models for each patient would be impractical. Our method offers a more feasible and scalable path toward real-world neurotechnology deployment.

---

> > > > ### Comment · Reviewer_bWGu · 2025-08-08
> > > >
> > > > The author's response has addressed my concerns, and I will increase my rating.

---

> > > > > ### Author Response · Authors · 2025-08-08
> > > > >
> > > > > Thank you for the updated rating. We truly appreciate it.

---

### Note · Authors · 2025-08-13

**We sincerely thank all reviewers and the AC for their constructive engagement throughout the review process. We are encouraged that our rebuttal and follow-up clarifications addressed the majority of concerns, as reflected in the multiple score increases and positive comments.**

Our work introduces ZEBRA, **the first framework for zero-shot fMRI-to-image decoding**. Unlike prior subject-specific pipelines, ZEBRA operates in a subject-agnostic setting without per-subject fine-tuning, making it significantly more scalable and practical for real-world brain–computer interface (BCI) applications. We believe this contribution is both novel and impactful: **zero-shot decoding is a critical step toward translating research-grade brain decoding models into deployable neurotechnology.**

We appreciate the opportunity to clarify the scope of meaningful applications. Our approach can benefit multiple domains, including assistive communication for patients with aphasia or locked-in syndrome, rehabilitation via visual neurofeedback in motor recovery, mental health monitoring (e.g., tracking depressive states), and sleep and dream research by decoding visual content during sleep.
The zero-shot nature of ZEBRA allows deployment even in resource-limited clinical settings without AI experts or expensive per-patient retraining—an important step toward wider use of neurotechnology.

We thank the reviewers for recognizing the novelty, careful design choices aligned with relevant brain areas, and the thorough additional experiments we provided during the discussion phase. We believe ZEBRA establishes a new benchmark and direction for subject-agnostic brain decoding, and we hope the AC will view its contributions as a strong step forward for both computational neuroscience and real-world BCI deployment.

---

### Decision · Program_Chairs · 2025-09-17

**Decision:**

Accept (poster)

**Comment:**

This paper received overall positive reviews. Reviewers acknowledged that it addresses a critical challenge in fMRI-to-image neural decoding and provides thorough experimental validation. The rebuttal effectively resolved initial concerns, including those related to novelty and missing comparisons, clarifying the paper to an acceptance-worthy level. The AC concurs with the majority and considers this a solid contribution. It is recommended that the clarifications and supporting evidence from the rebuttal be incorporated into the camera-ready version. We look forward to seeing this work presented at NeurIPS.